# Comparison of Ameliorative Effects between Probiotic and Biodegradable *Bacillus subtilis* on Zearalenone Toxicosis in Gilts

**DOI:** 10.3390/toxins13120882

**Published:** 2021-12-10

**Authors:** Wenqiang Shen, Yaojun Liu, Xinyue Zhang, Xiong Zhang, Xiaoping Rong, Lihong Zhao, Cheng Ji, Yuanpei Lei, Fengjuan Li, Jing Chen, Qiugang Ma

**Affiliations:** 1State Key Laboratory of Animal Nutrition, China Agricultural University, Beijing 100193, China; Shenwenqiang1996@163.com (W.S.); yaojunliu@outlook.com (Y.L.); zhangxinyue090@163.com (X.Z.); Zhangxiongcau@163.com (X.Z.); rxp409291@163.com (X.R.); zhaolihongcau@cau.edu.cn (L.Z.); jicheng@cau.edu.cn (C.J.); lypcau@126.com (Y.L.); 2FuQing Fengze Agricultural Science and Technology Development Co., Ltd., Fuzhou 350011, China; lifengjuan-1986@163.com (F.L.); chenjing101488@163.com (J.C.)

**Keywords:** zearalenone, degradable *Bacillus subtilis*, probiotic *Bacillus subtilis*, gilts

## Abstract

This study was conducted to compare the potential ameliorative effects between probiotic *Bacillus subtilis* and biodegradable *Bacillus subtilis* on zearalenone (ZEN) toxicosis in gilts. Thirty-six Landrace×Yorkshire gilts (average BW = 64 kg) were randomly divided into four groups: (1) Normal control diet group (NC) fed the basal diet containing few ZEN (17.5 μg/kg); (2) ZEN contaminated group (ZC) fed the contaminated diet containing an exceeded limit dose of ZEN (about 300 μg/kg); (3) Probiotic agent group (PB) fed the ZC diet with added 5 × 10^9^ CFU/kg of probiotic *Bacillus subtilis* ANSB010; (4) Biodegradable agent group (DA) fed the ZC diet with added 5 × 10^9^ CFU/kg of biodegradable *Bacillus subtilis* ANSB01G. Results showed that *Bacillus subtilis* ANSB010 and ANSB01G isolated from broiler intestinal chyme had similar inhibitory activities against common pathogenic bacteria. In addition, the feed conversion ratio and the vulva size in DA group were significantly lower than ZC group (*p* < 0.05). The levels of IgG, IgM, IL-2 and TNFα in the ZC group were significantly higher than PB and DA groups (*p* < 0.05). The levels of estradiol and prolactin in the ZC group was significantly higher than those of the NC and DA groups (*p* < 0.05). Additionally, the residual ZEN in the feces of the ZC and PB groups were higher than those of the NC and DA groups (*p* < 0.05). In summary, the ZEN-contaminated diet had a damaging impact on growth performance, plasma immune function and hormone secretion of gilts. Although probiotic and biodegradable *Bacillus subtilis* have similar antimicrobial capacities, only biodegradable *Bacillus subtilis* could eliminate these negative effects through its biodegradable property to ZEN.

## 1. Introduction

Zearalenone (ZEN), known as an F-2 mycotoxin, is a powerful estrogenic metabolite produced by certain species of *Fusarium* and *Erysipelas* spp. [1]. Several results showed that many feedstuffs for animals have been seriously contaminated with ZEN around the world [2,3], which could cause hyperestrogenism and fertility disorder in sows [4]. Moreover, a previous study found that exposure of post-weaning gilts to ZEN could increase the oxidative stress and had a negative impact on genital organs [5]. In addition, recent study found that ZEN can interfere with immune mediators at the spleen level and induce an intense inflammatory response [6]. Therefore, it is critical to find the appropriate and effective detoxifying strategies to prevent contamination by ZEN in animal husbandry. 

Numerous researches have reported that physical, chemical and biological methods can be used to counter mycotoxicosis [7,8,9]. However, most of these methods are impractical or potentially unsafe because of losses in the nutritional value, high equipment costs and formation of toxic residues or derivatives [10]. Biodegradation is eco-friendly and highly efficient in minimizing the harmfulness of mycotoxins in feeds [11,12]. Previous work from our laboratory reported that *Bacillus subtilis* ANSB01G, which has both a biodegradable effect against ZEN and probiotic activities against pathogenic bacteria, can alleviate toxicosis of ZEN in pre-pubertal female gilts [13,14]. However, the article did not clarify whether the reduced toxicity was due to its probiotic or biodegradable properties.

Therefore, the aim of this study was to investigate the effects of biodegradable and probiotic *Bacillus subtilis* on growth performance, serum biochemical indexes and hormone, serum antioxidant, immune indicators and mycotoxin residue in gilts exposed to ZEN for 25 d in vivo, as well as the inhibitory activity of common harmful bacteria in vitro.

## 2. Results

### 2.1. Biochemical and Physiological Characteristics of Bacillus subtilis ANSB010 and ANSB01G

The colony morphologies showed that the surfaces of *Bacillus subtilis* ANSB010 and ANSB01G colonies are rough, opaque and milky white (Figure 1A,B). Under the microscope, the cells were found to be short, thin rods, positive for Gram staining and capable of forming spores (Figure 1C,D). As shown in Table 1, the physiological and biochemical results revealed ANSB010 and ANSB01G had typical characteristics of *Bacillus* spp., such as growing well at 37 °C but not at 10 and 55 °C; and being able to utilize cellulose, glucose and maltose as the only carbon source. Moreover, a phylogenetic tree based on 16s rDNA sequences suggested that both ANSB010 and ANSB01G have a close evolutionary relationship to *Bacillus subtilis* (Figure 1E).

### 2.2. Bacteriostatic and ZEN-Degrading Effects of ANSB010 and ANSB01G

As shown in Figure 2 and Table 2, probiotic *Bacillus subtilis* ANSB010 and biodegradable *Bacillus subtilis* ANSB01G have a visible antibacterial effect on *Escherichia coli* (*E. coli*), *Salmonella choleraesuis* (*S. choleraesuis*) and *Staphylococcus aureus* (*S. aureus*) compared to the control group (Con) (*p* < 0.05, Table 2), while there was no significant difference in the antibacterial effect between ANSB010 and ANSB01G (*p* > 0.05, Table 2). Importantly, we noticed that ANSB01G could degrade 65.13%, 92.57% and 100.00% of ZEN in the fermentation broth at 6 h, 24 h and 48 h, respectively, but ANSB010 could not (*p* < 0.05, Table 3). Additionally, Appendix A show the representative chromatograms of degradation tests.

### 2.3. Growing Performance

As shown in Table 4, no significant difference were observed for the initial weight, terminal weight and average daily gain (ADG) (*p* > 0.05), while there was a decreasing trend of average daily feed intake (ADFI) in probiotic *Bacillus subtilis* ANSB010 agent (PB) and biodegradable *Bacillus subtilis* ANSB01G agent (DA) groups (*p* = 0.10) compared to ZEN contaminated (ZC) group. Of the four groups, the ZC group had the highest feed conversion ratio (F/G) value (*p* < 0.05), and was 1.07- and 1.13-fold higher than the PB and DA group, respectively.

### 2.4. Vulva Size

The effects of the four diets on the vulva size are shown in Table 5. The vaginal length and area of the DA group was significantly lower compared to the ZC group (*p* < 0.05), and there was no significant difference between the ZC and PB groups (*p* > 0.05). Interestingly, there was a decreasing trend of vaginal width among four groups (*p* = 0.10). The vaginal volume of the DA group was dramatically lower compared to all other groups (*p* < 0.05), and that of the PB group was significantly lower than that of the ZC group (*p* < 0.05), but no significant difference existed between the ZC and normal control diet (NC) groups, or PB and NC groups (*p* > 0.05). There was no significant difference in the vaginal height among the four groups (*p* > 0.05).

### 2.5. Serum Biochemical Indicators, Antioxidant and Immunology Parameters

There was no significant difference in the serum biochemical indicators (e.g., total protein (TP), albumin (ALB), alkaline phosphatase (ALP), aspartate aminotransferase (AST), alanine aminotransferase (ALT), creatinine (CRE) and urea nitrogen (BUN)) among these groups (Figure 3A–G, *p* > 0.05), nor in the levels of glutathione peroxidase (GSH-Px) (Figure 4A, *p* > 0.05) and malondialdehyde (MDA) (Figure 4C, *p* > 0.05), which were indicators of antioxidant activities. Interestingly, the level of superoxide dismutase (SOD) in the PB group was significantly lower than that of the ZC group (Figure 4B, *p* < 0.05).

As shown in Figure 5, the levels of immunoglobulin G (IgG) and IgM in the PB and DA groups were dramatically lower than that of the ZC group (Figure 5B,C, *p* < 0.05). However, there was no significant difference in the levels of IgA (Figure 5A, *p* < 0.05). In addition, the levels of pro-inflammatory factors (e.g., interleukin 2 (IL-2) and tumor necrosis factor-α (TNFα)) in the serum of the ZC group was significantly higher than these of other groups (Figure 5E,G, *p* < 0.05). Additionally, the DA group had the lowest level of IL-2 (Figure 5E, *p* < 0.05). However, other pro-inflammatory factors (e.g., IL-1β and IL-6) were not significantly different (Figure 5D,F, *p* > 0.05). 

### 2.6. Serum Hormone Parameters

The effects of diet supplemented with ZEN or *Bacillus subtilis* on the serum hormone of gilts were also shown in Figure 6. The level of estradiol (E2) in the NC and DA groups were significantly lower than that in the ZC groups (Figure 6C, *p* < 0.05), while no significant difference in the levels of follicle-stimulating hormone (FSH) and luteinizing hormone (LH) were observed in each treatment group (Figure 6A,B, *p* > 0.05). In addition, the level of prolactin (PRL) in the ZC group was the highest among the four groups (Figure 6D, *p* < 0.05). There was no difference in the PRL levels of the PB and DA groups compared to the NC group (Figure 6D, *p* > 0.05).

### 2.7. ZEN Residues

As shown in Table 6, mildewed maize enormously increased the content of ZEN in feed and feces. The NC group had very low levels of ZEN in feed (*p* = 0.06) and fecal samples (*p* < 0.05). In contrast, the ZC group had the highest levels of ZEN in feed and fecal samples (*p* < 0.05). Surprisingly, the content of ZEN in feces of the DA group were dramatically lower than the ZC and PB groups (*p* < 0.05). Additionally, the ZEN content in feces of the PB group was nearly same to that of the ZC group (*p* > 0.05). Intriguingly, although the ratio of ZEN contents between feces to diet was not significantly among the four groups (*p* > 0.05), the value in the DA group was indeed half that of the remaining three groups. In this study, both ZEN and its metabolites (α-zearalanol (α-ZAL), β-zearalanol (β-ZAL), α-zearalenol (α-ZOL), β-zearalenol (β-ZOL) and zearalanone (ZAN)) were not found in serum samples.

## 3. Discussion

The recent years have witnessed growing interests in finding practical and effective methods to detoxify ZEN in contaminated cereals and feeds [15,16,17]. Previous studies had been focused on mycotoxin adsorbents used to control mycotoxins in animal feed [18]. However, these adsorbents would contribute to environmental pollution as they transfer mycotoxins to surrounding areas [19]. Biodegradation of mycotoxins was considered as an efficient and environmentally protective method for the treatment of contaminated diets in the livestock production [11]. Previous studies have shown that some strains of *Bacillus* spp. were able to prevent the toxicity of ZEN [13,14], while it is not well known whether these effects were due to their probiotic or degradative capacities. Previous reports from our laboratory suggested biodegradable *Bacillus subtilis* ANSB01G could degrade 84.58%, 83.04% and 66.34% of ZEN in naturally contaminated maize, swine complete feed and dried distillers’ grains with solubles, respectively [13]. For these reasons, it is worth comparing the ameliorative effects between probiotic and biodegradable *Bacillus subtilis* in a ZEN-contaminated diet. The persuasion of the present study was ensured by the similarity of *Bacillus subtilis* ANSB010 and ANSB01G on the habitat source and the bacteriostatic activity against common pathogenic bacteria, including *E. coli, S. choleraesuis* and *S. aureus*. 

It has been reported that the presence of ZEN reduced feed consumption, caused a subsequent growth depression, and increased susceptibility to diseases [20,21]. Although we did not observe significant difference in gilts weight in the present study, there is a decreasing trend in ADFI among these groups. In line with previous reports [22], our study showed that the gilts fed on diets containing ZEN significantly increased F/G. In our study, the contaminated diet contained about 300 μg/kg of ZEN, leading to an increase in ZEN levels in the feces. Our data revealed that biodegradable *Bacillus subtilis* ANSB01G alleviated the toxicity while significantly decreasing the F/G, but probiotic *Bacillus subtilis* ANSB010 did not. 

Previous study has indicated that vulva swelling is the main clinical symptom of ZEN-induced toxicity in mammals [23,24], which has an adverse impact on the reproductive system and the breeding performance. Similarly, our data also indicated that ZEN significantly caused an increase in vulva size. Previous research found that the mycotoxin biodegradation agent composed of *Bacillus subtilis* ANSB01G and *Devosia* sp. ANSB714 can effectively reduce the estrogenic swelling of the vulva caused by ZEN in immature gilts [25]. In this study, we also discovered that only biodegradable *Bacillus subtilis* ANSB01G mediates vulva swelling caused by ZEN. Some available evidence also demonstrated the obvious adverse effects of ZEN on the secretion of these hormones and productivity of animals [26]. In addition, similar results were reflected by a fluctuation in hormone levels. Although there were no significant difference in the levels of FSH and LH, ZEN diets markedly increased the level of E2 in the gilts. It was well known that ZEN is a competitive substrate for endogenous estrogens, binding to estrogen receptors and thereby having a damaging effect on the function of gonads [27]. A recent long-term (48 d) study found that low doses of ZEN (20 μg/kg BW) induced changes in the concentrations of E2 levels in pre-pubertal gilts [28]. The addition of biodegradable *Bacillus subtilis* ANSB01G, in this study, restored serum E2 levels and modulated the function of gonads in gilts. 

Van and his colleagues reported that ZEN ingestion partially induced oxidative stress in piglets, as it revealed an increased content of MDA and activity of SOD [29]. In addition, even low levels of ZEN (246 ug/kg) in the diet of gestation sows can lead to an increase in the level of serum MDA and cause cell apoptosis and moderate lesions of the liver, kidney, uterus, and ovary [30]. Importantly, our research showed that only biodegradable *Bacillus subtilis* ANSB01G reversed these increasing trends. However, no changes in the levels of antioxidant enzymes (e.g., GSH-Px and MDA) or SOD activity were observed in the serum of gilts. Moreover, we did not observe that diets treated with ZEN or both strains of *Bacillus subtilis* affected the serum biochemical indicators (e.g., ALB, AST, ALP and CRE). These data were inconsistent with the previous reports [31,32]. The difference in results might be attributed to the age difference of the animal model and different doses of ZEN contaminated. However, we noticed that ZEN increased IgG level in serum, but not in the levels of IgA and IgM. Gilts treated with biodegradable *Bacillus subtilis* ANSB01G and probiotic *Bacillus subtilis* ANSB010 decreased IgG levels in serum. IgG, IgM and IgA are the main components of immunoglobulins, of which the content of IgG is up to 70–75%. These results suggested that ZEN has antigenic activity which stimulated the immune system of gilts, and both biodegradable and probiotic *Bacillus subtilis* have the capacity to recover the stimulation caused by ZEN. 

Moreover, it has been reported that ZEN could increase the synthesis and expression of pro-inflammatory factors through JNK signaling pathway activation [6]. Currently, few studies have focused on the effects of ZEN on the modulation of inflammation in gilts. In the present study, ZEN in the feed significantly increased the levels of TNFα and IL-2, and had no effects on IL-1β and IL-6 in serum. An increase in levels of TNFα, one of the most powerful pro-inflammation factors, might generate a risk of a more severe inflammatory response [33]. The inflammatory response in this trial was similar to previous results from our lab that showed an enormous increase in inflammatory cytokines (e.g., IL-2, IL-8 and IL-10) [25]. The elevation of cytokines caused by ZEN would impair the erythroid progenitor and red blood cells [34], which revealed a potential cancerotoxic effect of ZEN. Biodegradable *Bacillus subtilis* ANSB01G mitigated the acute inflammatory response, confirming the reliability of a biodegradation approach in myotoxin degradation. In comparison, the lack effect of probiotic *Bacillus subtilis* ANSB010 on the elevated inflammatory response demonstrated that probiotic *Bacillus subtilis* had no effects on alleviating the toxicity of ZEN. Therefore, we concluded that biodegradable *Bacillus subtilis* was more protective against ZEN toxicosis in gilts than the probiotic *Bacillus subtilis*. Taken together, this study provided further evidence that the specific strain of *Bacillus subtilis* ANSB01G can alleviate the toxicity of ZEN, mainly due to its biodegradable capacity.

## 4. Conclusions

This study demonstrated that a feeding diet contaminated with ZEN of 300 μg/kg had a damaging impact on the growth performance, plasma immune function and hormone secretion of gilts. Although probiotic *Bacillus subtilis* ANSB010 and biodegradable *Bacillus subtilis* ANSB01G have similar antimicrobial capacities and alleviate inflammatory responses, only biodegradable *Bacillus subtilis* ANSB010 could regulate estrogen levels, relieve swelling of the vulva, and reduce the F/G and fecal ZEN residues. Hence, the biodegradable *Bacillus subtilis* ANSB01G used in this study is considered to have great and promising potential for biodegradation of mycotoxin in feed industrial applications. 

## 5. Materials and Methods

### 5.1. Source and Identification of Bacterial Strains

The two strains of *Bacillus subtilis* used in this study were isolated from healthy broiler intestinal chyme and identified and characterized using a standard method described by Holt et al. [35]. Gram-staining was performed using the Gram staining kit (G1060, Solarbio, Beijing, China). Briefly, the bacteria were activated in Luria-Bertani (LB) medium at 37 °C for 12 h. Then, a 2.5 μL sample was stained with crystal violet for 1 min, mordanted with iodine solution 1 min, decolorized for 30 s, and counterstained with safranine for 1 min. Spore-staining was performed using the Spore stain kit (G1133, Solarbio, Beijing, China). First, a 5 μL sample, cultured in LB medium at 37 °C for 48 h, was stained in malachite green solution for 10 min and counterstained with safranine for 3 min. Finally, the staining results were observed using a microscope. Then, DNA was extracted from the bacterial isolates using the Bacterial Genomic DNA kit (Beijing Zoman Biotechnology Co., Ltd., Beijing, China) according to the manufacturer’s instructions. PCR amplification of 16S rDNA was performed with the primers 27F (5′-AGAGTTTGATCMTGGCTCAG-3′) and 1492R (5′-CGGTTACCTTGTTACGACTT-3′), and PCR products were purified and sequenced by Sangon Biotech (Beijing, China). Probiotic and biodegradable *Bacillus subtilis* were identified as strains of *Bacillus subtilis* named ANSB010 and ANSB01G, respectively. It has been shown that ANSB01G could degrade ZEN in naturally contaminated maize with high efficiency [13]. The *Bacillus subtilis* ANSB01G in our experiment was a domesticated strain based on the wild bacteria obtained by Lei et al. [13]. The method of microbial domestication was as follows: after activation, *Bacillus subtilis* ANSB01G was induced and cultured in a series of MRS mediums with gradually increasing concentrations of ZEN [36]. The efficiency of *Bacillus subtilis* ANSB01G in degrading ZEN was further improved after several domestications. The 16 s rDNA sequences of ANSB010 and domesticated ANSB01G are shown in the Appendix A. A phylogenetic tree was drawn with the neighbor-joining method of 1000 bootstrap replications within Mega 5.0. 

### 5.2. Antibacterial Activity and ZEN Degradation Tests

Selected indicated bacteria *E. coli* (No.10003), *S. choleraesuis* (No.21493) and *S. aureus* (No.10384) were purchased from the China Center of Industrial Culture Collection (CICC, Beijing, China). *E. coli, S. choleraesuis* and *S. aureus* were inoculated in MRS medium and incubated in MRS medium at 180 rpm for 24 h at 37 °C. After diluting twice in a gradient, 200 μL of the bacterial solution was added to MRS medium and spread evenly, then z sterilized Oxford cup was placed on the medium. Then, 200 μL of the supernatant of the probiotic and biodegradable *Bacillus subtilis* solution was aspirated into an Oxford cup and the non-inoculated MRS medium was used as a control. After incubating at 37 °C for 24 h, the antibacterial circle diameter (cm) was measured. For the ZEN degradation test, ZEN solution (100 μL, 2 μg/mL) was added into the LB medium of ANSB010 and ANSB01G (900 μL, 3.0 × 10^8^ CFU/mL) and incubated with shaking for 6 h, 24 h, and 48 h at 37 °C in the dark, followed by measurement of ZEN levels using the HPLC method.

### 5.3. Animals and Experimental Treatments

The animal experiments were conducted according to the animal welfare requirements and approved by the Animal Protocol Review Committee of the China Agriculture University (Beijing, China).

Thirty-six healthy gilts (Landrace × Yorkshire, average body weight = 64 kg) were selected for the experiment. Then, these animals were randomly assigned to four treatments with nine replicates of one gilt per replicate for each group: (1) Normal control diet group (NC) fed the basal diet containing few ZEN (17.5 μg/kg diet) by controlling the quality of maize; (2) ZEN-contaminated group (ZC) fed the contaminated diet containing an exceeded limit dose of ZEN (about 300 μg/kg diet) by replacing normal maize in the basal diet with moldy maize; (3) Probiotic agent group (PB) fed the ZC diet with added 5 × 10^9^ CFU/kg of probiotic *Bacillus subtilis* ANSB010; (4) Biodegradable agent group (DA) fed the ZC diet with added 5 × 10^9^ CFU/kg of biodegradable *Bacillus subtilis* ANSB01G. The contaminated maize was purchased in 2018 from a small family farm in Henan Province, China. The *Bacillus subtilis* ANSB010 and ANSB01G were incubated in LB medium for 24 h at 37 °C, followed by drying at 65 °C, and then evenly mixed into the diets. Diets are formulated to meet or exceed nutrient requirements (Table 7) recommended by the National Research Council for replacement gilts (NRC, 2012). The experimental period lasts for 25 d. The contaminated diets were prepared by replacing corn in the control with the naturally contaminated maize. During the supplementation period, all piglets were individually housed in temperature-controlled stainless steel metabolism pens (25 ± 2 °C), allowing free access to drinking water. Animal care and experimental procedures were in accordance with the guidelines of the National Institutes of Health Guide and the China Ministry of Agriculture for the care and use of laboratory animals.

### 5.4. Growth Performances

The initial body weight and terminal body weight were recorded. Moreover, ADG, ADFI, and F/G (ADFI/ADG) were calculated.

### 5.5. Vulva Size Determination

The vulva length (a), width (b) and height (h) were measured and recorded at 0, 12d and 24 d. The determination of vulva area and volume was performed and results were calculated according to the method previously described by Zhao and his colleagues [14]. The area and volume of vulva are approximately elliptical and conical, respectively. Therefore, the area of the vulva is in accordance with the formula: S = (π × a × b)/4; and the volume of the vulva is calculated with the equation: V = 1/3 × S × h. 

### 5.6. Serum Parametes

Blood was collected from the marginal ear vein at the end of the experiment period. Then, the blood samples were centrifuged at 3000× *g* for 10 min to obtain serum for further analysis. The serum biochemical indicators including the TP, ALB, ALT, AST and ALP, CRE and BUN were measured with an automatic biochemical analyzer (Hitachi 7160, Hitachi High-Technologies Corporation, Tokyo, Japan). Immunoglobulin A (IgA), IgG and IgM were also determined with an automatic biochemical analyzer (Hitachi 7160). In addition, interleukin 1β (IL-1β), IL-2, IL-6 and TNFα were measured by an enzyme-linked immune sorbent assay (ELISA) kit (YuanMu Biotechnology, Shanghai, China). Serum SOD, GSH-Px and MDA were detected according to the instructions of the manufacturer using microplate test kits (Nanjing Jiancheng Bioengineering Institute, Nanjing, China). The concentrations of serum E2, FSH, LH and PRL were determined by radioimmunoassay (RIA). The serum samples were treated with radioactive-125I according to the instructions of the RIA kit (Beijing Kemei Biotechnology Co., Ltd., Beijing, China). Then, a GC-1200 radio-immunity gamma-counter (KeDa Innovation Co., Ltd., Hefei, China) was used to measure hormone concentrations. For each RIA, the intra- and inter-assay coefficients of variation were less than 15% and less than 10%, respectively. Hormone concentration was determined according to each sample’s level of radioactivity.

### 5.7. Determination of ZEN in the Feed, Feces, Broth and Serum 

Two days before the end of the test, fresh fecal samples were taken to determine the content of ZEN in feces according to the description of Chinese certification GB/T 23504-2009 and Lei et al. [13]. For feed and feces samples, 50 g ground samples were extracted by acetonitrile-water (70:30, *v/v*, 200 mL), followed by filtration with Whatman 4 filter paper. After dilution with PBS solution (PH = 7.40), the mixing solution were filtered through a micro-filter. A volume of 20 mL of suspension was passed through the immunoaffinity column (Femdetection FD-C21, Nanjing, China) at a flow rate of 1.0 mL/min under gravity. After washing the column with distilled water, the ZEN was subsequently eluted with 2 mL of methanol into a centrifuge tube for HPLC analysis. For liquid medium, samples (1 mL) were extracted with acetonitrile (9 mL) at 180 rpm for 2 h. The mixed samples were filtered using glass fibre filter paper, and then collected for subsequent HPLC analysis. ZEN and its metabolites in serum were analyzed using the method described by Duca et al. [37]. Briefly, serum (2 mL) was mixed with buffer ammonium acetate solution (8 mL). The mixed solution was incubated with glucuronidase/arylsulfatase (50 μL) at 37 °C for 15 h. After the samples were centrifuged at 5000 rpm for 10 min, the supernatant was passed through the immunoaffinity column. The column was then rinsed with 20 mL of ultrapure water. The analytes were then eluted with acetonitrile (2 mL). Subsequently, the solution was dried using a Speed Vac concentration system after which 200 μL mobile phase was added. For HPLC analysis, 20 μL sample solution was injected into the HPLC system. Separation was in a C18 column (4.6 mm × 150 mm, 5 μm; Thermo Fisher Scientific, Waltham, MA, USA) with mobile phase (water: acetonitrile, 50:50, *v/v*) at a flow rate of 1.0 mL/min. The analytes were detected by a fluorescence detector (Waters, Milford, MA, USA), excitation and emission wavelengths were 274 and 440 nm, respectively. The retention time was 7–8 min.

### 5.8. Statistical Analysis

Data were analyzed statistically using SAS 9.4 software (SAS Institute Inc., Cary, NC, USA) and were presented as mean ± SEM. The significance of difference between groups of gilts were analyzed by one-way ANOVA. A normality test (Shapiro–Wilk) was performed to determine normality before one-way ANOVA analysis. Differences were regarded as statistically significant at a probability of *p* < 0.05, and *p*-values < 0.10 were regard as a trend. 

## Figures and Tables

**Figure 1 toxins-13-00882-f001:**
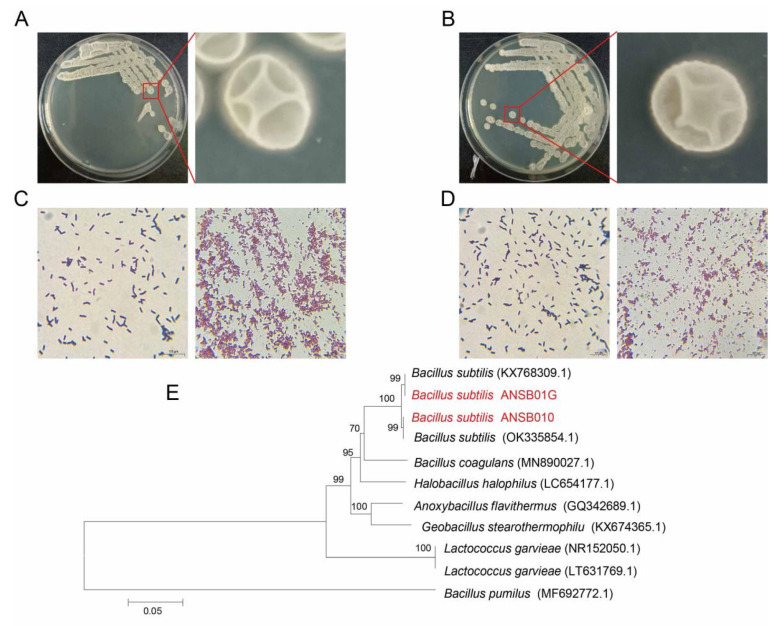
Colony characteristics of ANSB010 (**A**) and ANSB01G (**B**); cell (**left**) and spore (**right**) morphology of ANSB010 (**C**) and ANSB01G (**D**), scale bar, 100 μm; (**E**) the phylogenetic tree of *Bacillus subtilis* ANSB010 and ANSB01G, the GenBank accession numbers of sequences are shown in round brackets.

**Figure 2 toxins-13-00882-f002:**
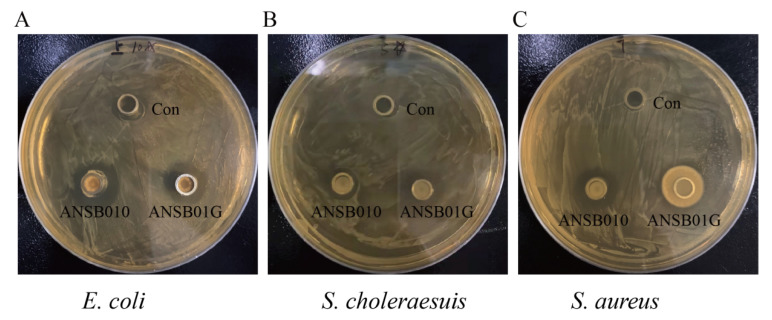
The antibacterial effects of probiotic *Bacillus subtilis* ANSB010 and biodegradable *Bacillus subtilis* ANSB01G on *E. coli* (**A**), *S. choleraesuis* (**B**) and *S. aureus* (C). *E. coli*, *Escherichia coli*; *S**. choleraesuis*, *Salmonella choleraesuis*; *S. aureus*, Staphylococcus aureus. Con: MRS medium; ANSB010, probiotic *Bacillus subtilis* ANSB010; ANSB01G, biodegradable *Bacillus subtilis* ANSB01G.

**Figure 3 toxins-13-00882-f003:**
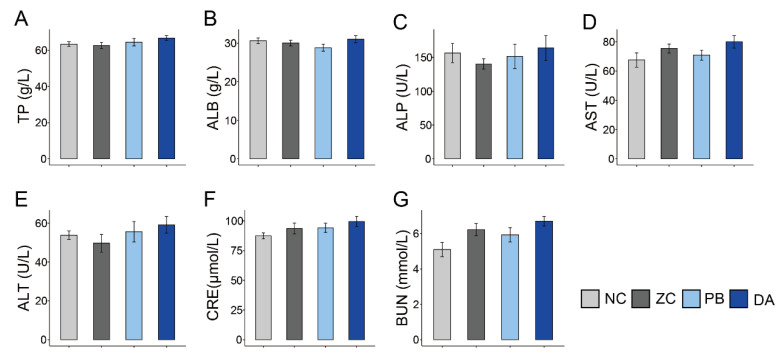
Comparison of serum levels of biochemical indicators among different treatment groups. (**A**) TP, total protein; (**B**) ALB, albumin; (**C**) ALP, alkaline phosphatase; (**D**) AST, aspartate aminotransferase; (**E**) ALT, alanine aminotransferase; (**F**) CRE, creatinine; (**G**) BUN, urea nitrogen. NC, normal control diet group; ZC, ZEN-contaminated group; PB, probiotic *Bacillus subtilis* group; DA, biodegradable *Bacillus subtilis* group.

**Figure 4 toxins-13-00882-f004:**
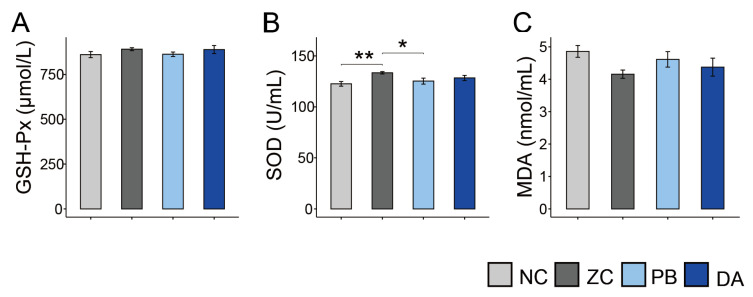
Comparison of serum levels of antioxidant parameters among different treatment groups. (**A**) GSH-Px, glutathione peroxidase; (**B**) SOD, superoxide dismutase; (**C**) MDA, malondialdehyde. NC, normal control diet group; ZC, ZEN-contaminated group; PB, probiotic *Bacillus subtilis* group; DA, biodegradable *Bacillus subtilis* group. *, *p* < 0.05; **, *p* < 0.01.

**Figure 5 toxins-13-00882-f005:**
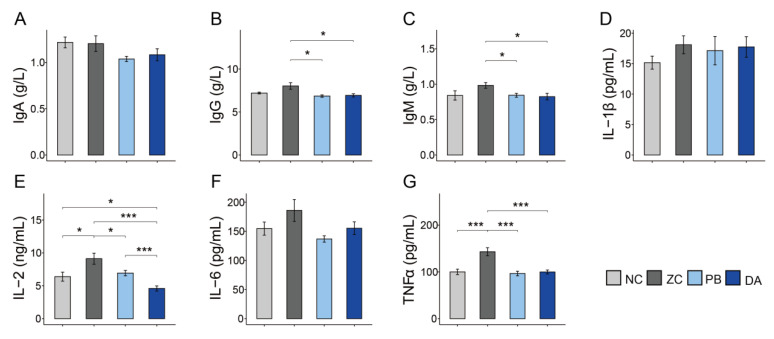
Comparison of serum levels of immune and inflammatory parameters in different treatment groups. (**A**–**C**) IgA, immunoglobulin A; IgG, immunoglobulin G; IgM, immunoglobulin M; (**D**–**G**) IL-1β, interleukin 1β; IL-2, interleukin 2; IL-6, interleukin 6; TNFα, tumor necrosis factor-α; NC, normal control diet group; ZC, ZEN-contaminated group; PB, probiotic *Bacillus subtilis* group; DA, biodegradable *Bacillus subtilis* group. *, *p* < 0.05; ***, *p* < 0.001.

**Figure 6 toxins-13-00882-f006:**
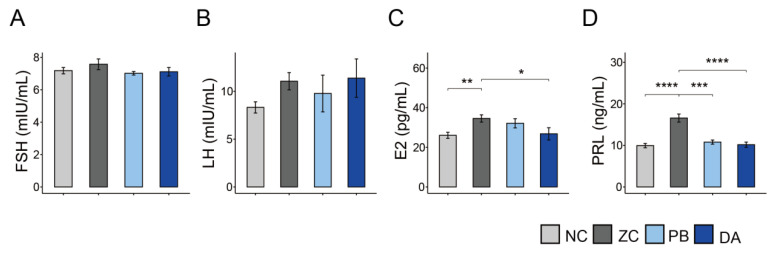
Comparison of serum levels of hormone parameters in different treatment groups. (**A**–**D**) FSH, follicle-stimulating hormone; LH, luteinizing hormone; E2, estradiol; PRL, prolactin. NC, normal control diet group; ZC, ZEN-contaminated group; PB, probiotic *Bacillus subtilis* group; DA, biodegradable *Bacillus subtilis* group. *, *p* < 0.05; **, *p* < 0.01; ***, *p* < 0.001; ****, *p* < 0.0001.

**Table 1 toxins-13-00882-t001:** Biochemical and physiological characteristics of *Bacillus subtilis* ANSB010 and ANSB01G.

Experimental Projects	ANSB010 ^1^	ANSB01G ^2^	Experimental Projects	ANSB010	ANSB01G
Gram	+	+	Glucose	+	+
Cell shape	Rod-shape	Rod-shape	Maltose	+	+
Cell diameter > 1 μm	+	+	Sucrose	+	+
Spore forming	+	+	D-xylose	+	+
Spore dilation	−	−	L-xylose	−	−
Round spores	−	−	D-arabinose	−	−
Glycerol	+	+	L-arabinose	+	+
Cellulose utilization	+	+	D-mannitol	+	+
Catalase	+	+	Gas production using glucose	−	−
Oxidase test	+	+	Citrate utilization	+	+
Anaerobic	−	−	Growth at 10 or 50 °C	−	−
Voges-Proskauer (VP) test	+	+	Growth at 37 °C	+	+
VP < pH 6	+	+	Growth at pH 5.7	+	+
VP > pH 7	−	−	Growth on 7% NaCl	+	+
Methyl red test	−	−	Hydrolysis of starch	+	+
Gluconate	−	−	Decomposition of casein	+	+
Xylitol	−	−	Nitrate reduction	+	+

^1^ ‘+’, ‘−’ and ‘w’ mean positive, negative and weak response, respectively. ^2^ The results of ANSB01G after domestication were consistent with Lei et al. [13].

**Table 2 toxins-13-00882-t002:** Antibacterial effects of *Bacillus subtilis* ANSB010 and ANSB01G.

Indicator Bacteria	Antibacterial Circle Diameter (cm)			SEM	*p*-Value
	Con	ANSB010	ANSB01G		
*E. coli*	1.10 ^b^	1.76 ^a^	1.68 ^a^	0.07	0.00
*S. choleraesuis*	0.86 ^b^	1.62 ^a^	1.71 ^a^	0.09	0.00
*S. aureus*	1.07 ^b^	1.74 ^a^	1.71 ^a^	0.08	0.00

Con: MRS medium; ANSB010, probiotic *Bacillus subtilis* ANSB010; ANSB01G, biodegradable *Bacillus subtilis* ANSB01G. Different superscript letters represent significant difference.

**Table 3 toxins-13-00882-t003:** The degradation rate (%) of ANSB010 and ANSB01G on zearalenone in fermentation medium.

Time	ANSB010	ANSB01G	SEM	*p*-Value
6 h	0.05 ^b^	65.13 ^a^	12.23	0.00
24 h	0.38 ^b^	92.57 ^a^	4.81	0.00
48 h	−0.26 ^b^	100.00 ^a^	4.22	0.00

ANSB010, probiotic *Bacillus subtilis* ANSB010; ANSB01G, biodegradable *Bacillus subtilis* ANSB01G. Different superscript letters represent significant difference.

**Table 4 toxins-13-00882-t004:** Comparison of growth performance among different treatment groups.

Items	NC	ZC	PB	DA	SEM	*p*-Value
Initial weight, kg	63.43	64.14	64.11	64.4	1.78	0.99
Terminal weight, kg	84.05	84.98	85.12	86.00	1.72	0.93
ADG, g	825	834	841	864	29.69	0.85
ADFI, g	2151	2244	2114	2047	81.44	0.10
F/G	2.61 ^abc^	2.69 ^ab^	2.52 ^bc^	2.38 ^c^	0.08	<0.01

ADG, average daily gain; ADFI, average daily feed intake; F/G, feed conversion ratio; NC, normal control diet group; ZC, Zearalenone (ZEN)-contaminated group; PB, probiotic *Bacillus subtilis* group; DA, biodegradable *Bacillus subtilis* group. Different superscript letters represent significant difference.

**Table 5 toxins-13-00882-t005:** The values of vulva indexes in different treatment groups.

Items	NC	ZC	PB	DA	SEM	*p*-Value
Length, cm	3.12 ^ab^	3.34 ^a^	2.93 ^ab^	2.85 ^b^	0.66	0.03
Width, cm	2.51	2.53	2.42	2.21	0.48	0.10
Height, cm	2.36	2.30	1.90	1.95	0.75	0.43
Area, cm^2^	6.15 ^ab^	6.64 ^a^	5.58 ^ab^	4.99 ^b^	0.21	0.03
Volume, cm^3^	4.83 ^ab^	5.11 ^a^	3.54 ^b^	3.23 ^c^	0.24	0.00

NC, normal control diet group; ZC, ZEN-contaminated group; PB, probiotic *Bacillus subtilis* group; DA, biodegradable *Bacillus subtilis* group. Different superscript letters represent significant difference.

**Table 6 toxins-13-00882-t006:** Content of ZEN in feed and fecal samples.

Items	NC	ZC	PB	DA	SEM	*p*-Value
Content of ZEN in diet, μg/kg	17.50	304.80	297.30	307.70	55.42	0.06
Content of ZEN in feces, μg/kg	7.23 ^b^	104.24 ^a^	98.86 ^a^	55.37 ^b^	17.34	0.00
Ratio of ZEN contents between feces to diet, %	41.34	34.20	33.26	17.30	5.87	0.73

NC, normal control diet group; ZC, ZEN-contaminated group; PB, probiotic *Bacillus subtilis* group; DA, biodegradable *Bacillus subtilis* group. Different superscript letters represent significant difference.

**Table 7 toxins-13-00882-t007:** Ingredients and compositions of the basal diet, as fed basis.

Ingredient	%	Nutrition Component	Content ^1^
Maize	57.00	DE, Kcal/Kg	3100
Soybean meal	23.00	Crude protein, %	17.00
Wheat bran	16.00	Calcium, %	0.76
Calcium hydrophosphate	1.00	Total phosphorus, %	0.61
Limestone	1.05	Non-phytate phosphorus, %	0.36
Salt	0.30	Lysine, %	0.85
Threonine	0.05	Methionine, %	0.52
Lysine 70%	0.60	Threonine, %	0.59
Choline chloride	0.12		
Chlortetracycline	0.05		
Compound-premix ^2^	0.83		
Total	100.00		

^1^ The value is calculated. ^2^ Supplied the following per kilogram of diet: vitamin A, 5000 IU; vitamin D3, 900 IU; vitamin E, 40 IU; vitamin K, 2.5 mg; vitamin B1, 1.5 mg; vitamin B2, 6.4 mg; vitamin B6, 2.5 mg; vitamin B12, 0.025 mg; pantothenate, 20 mg; nicotinic acid, 30 mg; choline, 0.50 g; Fe 100 mg; Cu, 6 mg; Zn, 50 mg; Mn, 20 mg; Se, 0.30 mg; I, 0.24 mg.

## Data Availability

Data are available from the first author.

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
