# Peer review of "Comparison of Ameliorative Effects between Probiotic and Biodegradable Bacillus subtilis on Zearalenone Toxicosis in Gilts"

_toxins, 2021, doi:10.3390/toxins13120882_

Round 1

Reviewer 1 Report

The paper entitled: Comparison of ameliorative effects between probiotic and biodegradable Bacillus subtilis on Zearalenone toxicosis in gilts, deals with the comparison of the potential ameliorative effects between probiotic Bacillus subtilis and biodegradable Bacillus subtilis on zearalenone toxicosis in gilts. It showed that zearalenone-contaminated diet had a damaging impact on growth performance, plasma immune function and hormones secretion of gilts. Although probiotic and biodegradable Bacillus subtilis have similar antimicrobial capacities, only biodegradable Bacillus subtilis could eliminate these negative effects through its biodegradable property to zearalenone. The alleviating effects of biodegradable Bacillus subtilis ANSB01G on the zearalenone poisoned gilts were compared to probiotic Bacillus subtilis ANSB010 isolated from the same source and with similar bacteriostatic activity. Results showed that the improvement of ANSB01G on zearalenone-poisoned gilts comes from its biodegradation activity to zearalenone, not from its antibacterial activity. The manuscript demonstrated that biodegradable Bacillus subtilis ANSB01G can be considered to have promising potential for biodegradation of mycotoxin in feed industrial applications. The paper is well written and can be followed with interest by the readers, therefore it can be published as it is. 

Author Response

Dear reviewer,

We greatly appreciate you for processing our paper. And many thanks for your affirmation of our article.

Enjoy your life and work!

Yours sincerely,

Reviewer 2 Report

The manuscript represents data on the evaluation of zearalenone-degrading strain of Bacillus subtilis in experiments in vitro and in vivo. Some issues authors should address before publishing the manuscript.

1) what is novel in strain identification compared to Lei, Y. P., Zhao, L. H., Ma, Q. G., Zhang, J. Y., Zhou, T., Gao, C. Q., & Ji, C. (2014). Degradation of zearalenone in swine feed and feed ingredients by Bacillus subtilis ANSB01G. World Mycotoxin Journal, 7(2), 143–151. doi:10.3920/wmj2013.1623

2) Please, describe the probe preparation from fermentation media and feed, for ZEA HPLC analysis. Please, indicate if the methods are original or published.

3) Did you evaluate data normality before ANOVA-tests?

4) Provide typical HPLC-chromatograms in the Supplement

5) provide the source, year of the contaminated corn

6) What a reason to evaluate ZEA in fecal masses only?

7) What a reason to spike the control feed with ZEA?

Author Response

Dear reviewer,

Thank you so much for your comments and advices. Those comments are all valuable and very helpful for revising and improving our paper, as well as the important guiding significance to our researches.

We have addressed comments carefully and have made correction which we hope meet with approval.

We have resubmitted a new version of the manuscript with this response. And the revisions are marked up using the “Track Changes” function.

Yours sincerely,

30th, November 2021

Round 2

Reviewer 2 Report

Thanks for the response. I have no further  comments.